# Using the Semantic Information G Measure to Explain and Extend Rate-Distortion Functions and Maximum Entropy Distributions

**DOI:** 10.3390/e23081050

**Published:** 2021-08-15

**Authors:** Chenguang Lu

**Affiliations:** 1School of Computer Engineering and Applied Mathematics, Changsha University, Changsha 410000, China; survival99@gmail.com; 2Institute of Intelligence Engineering and Mathematics, Liaoning Technical University, Fuxin 123000, China

**Keywords:** rate-distortion function, boltzmann distribution, semantic information measure, machine learning, maximum entropy, minimum mutual information, Bayes’ formula

## Abstract

In the rate-distortion function and the Maximum Entropy (ME) method, Minimum Mutual Information (MMI) distributions and ME distributions are expressed by Bayes-like formulas, including Negative Exponential Functions (NEFs) and partition functions. Why do these non-probability functions exist in Bayes-like formulas? On the other hand, the rate-distortion function has three disadvantages: (1) the distortion function is subjectively defined; (2) the definition of the distortion function between instances and labels is often difficult; (3) it cannot be used for data compression according to the labels’ semantic meanings. The author has proposed using the semantic information G measure with both statistical probability and logical probability before. We can now explain NEFs as truth functions, partition functions as logical probabilities, Bayes-like formulas as semantic Bayes’ formulas, MMI as Semantic Mutual Information (SMI), and ME as extreme ME minus SMI. In overcoming the above disadvantages, this paper sets up the relationship between truth functions and distortion functions, obtains truth functions from samples by machine learning, and constructs constraint conditions with truth functions to extend rate-distortion functions. Two examples are used to help readers understand the MMI iteration and to support the theoretical results. Using truth functions and the semantic information G measure, we can combine machine learning and data compression, including semantic compression. We need further studies to explore general data compression and recovery, according to the semantic meaning.

## 1. Introduction

Bayes’ formula is used for probability predictions. Using Bayes’ formula, from a joint (probability) distribution *P*(*x*, *y*), or distributions *P*(*x*|*y*) and *P*(*y*) (where *x* and *y* are variables), we can obtain the posterior distribution of *y*:(1)$$P(y|x)=\frac{P(x,y)}{\sum_{y}P(x,y)}=\frac{P(y)P(x|y)}{\sum_{y}P(y)P(x|y)}=\frac{P(y)P(x|y)}{P(x)}.$$

Similar expressions with Negative Exponential Functions (NEFs) and partition functions often appear in the rate-distortion theory [1,2,3], statistical mechanics, the maximum entropy method [4,5], and machine learning (see the Restricted Boltzmann Machine (RBM) [6,7] and the Softmax function [8,9]). For example, in the rate-distortion theory, the Minimum Mutual Information (MMI) distribution is:(2)$$P(y|x)=\frac{P(y)\exp[sd(x,y)]}{\sum_{y}P(y)\exp[sd(x,y)]}$$
where *s* is a negative parameter, and *d*(*x*, *y*) is the distortion function.

However, an NEF, such as exp[*sd*(*x*, *y*)], is not a (statistical) probability function because its sums, ∑*_x_* exp[*sd*(*x*, *y*)] and ∑*_y_* exp[*sd*(*x*, *y*)], are not 1. Its main feature is that its maximum value is exp(0) = 1. An NEF is more like a membership function [10], similarity function, or Distribution Constraint Function (DCF), which softly constrains a probability distribution. Why can we put an NEF as a probability function into a Bayes-like formula? Can we find a simple explanation for why these NEFs and partition functions exist in different areas? Although the relationship between the rate-distortion function and the maximum entropy method has been studied [11], we still need a more general probability theory or probability framework to explain the above phenomenon.

On the other hand, although the rate-distortion theory has achieved great successes [2,3,12,13,14], it still has the following disadvantages:Distortion function *d*(*x*, *y*) is subjectively defined, lacking the objective standard;It is hard to define the distortion function using distances when we use labels to replace instances in machine learning; where possible the labels’ number should be much less than the possible instances’ number. For example, we need to use “Light rain”, “Moderate rain”, “Heavy rain”, “Light to moderate rain”, etc., to replace daily precipitations in millimeters, or use “Child”, “Youth”, “Adult”, “Elder”, etc., to replace people’s ages. In these cases, it is not easy to construct distortion functions;We cannot apply the rate-distortion function for semantic compression, e.g., data compression according to labels’ semantic meanings.

The first two disadvantages remind us that we need to obtain allowable distortion functions from samples or sampling distributions by machine learning.

We consider the example of semantic compression as related to ages. For instance, *x* denotes an age (instance) and *y_j_* denotes the labels: *y*_1_ = “Non-adult”, *y*_2_ = “Youth”, *y*_3_ = “Adult”, and *y*_4_ = “Elder”. All *x* values that make *y_j_* true form a fuzzy set. The truth functions of these labels are also the membership functions of the fuzzy sets (see Figure 1). According to Davidson’s truth-conditional semantics [15], truth function *T*(*y_j_|x*) ascertains the semantic meaning of *y_j_*. For a given *P*(*x*) and the constraint, we need to find the Shannon channel *P*(*y*|*x*) that minimizes the mutual information between *y* and *x*. The Minimum Mutual Information (MMI) should be the lower limit of the average codeword length for coding *x* to *y*. The constraint condition states that labels’ selections should accord with the rules of the langauges used, expressed by the truth functions. In this case, the rate-distortion function cannot work well because it is not easy to define a distortion function which is compatible with the labels’ semantic meanings.

There have been meaningful studies on semantic compression [16,17,18]. However, in these studies, either the information-theoretic method related to the rate-distortion function or similar aspects, has not been adopted [18], or no semantic information measure has been used. The data compression or clustering related to perception has been studied in [19,20]; however, the discrimination functions (like the truth function) and the sensory information measure have not been adopted. For semantic compression, we need a proper information measure to measure semantic information and sensory information. We also want a function, like the rate-distortion function, related to labels’ semantic meanings.

To measure semantic information, researchers have proposed many semantic information measures [21,22,23,24,25] or information measures related to semantics [26,27]. However, it is not easy to use them for machine learning or semantic compression. For the similar purpose, the author proposed the semantic information G theory, or simply the G theory, in the 1990s [28,29,30]. The letter “G” means the generalization of Shannon’s information theory. The semantic information measure, e.g., the G measure, is defined with log(truth_function/logical_probability) or its average. The truth function and the logical probability are similar to the NEF and the partition function, respectively. The truth function can be expressed by not only the NEF, but also the Logistic function and other functions between 0 and 1. The semantic information G measure can also be used to measure semantic information conveyed not only by natural languages but also by sensory organs and measuring instruments, such as thermometers, scales, and GPS devices [31]. For sensory information, truth functions become discrimination functions or confusion probability functions [30].

The G measure measures labels’ semantic information only according to their extensions ascertained by truth functions, without considering their intentions. Therefore, we may call this semantic information, formally semantic information. For simplicity, this paper mainly considers the (formally) semantic information between label *y* and instance *x*. The semantic information between two labels or sentences is simply discussed in Section 6.4.

The G theory uses the P-T probability framework [32], which consists of both statistical probability (denoted by *P*) and logical probability (by *T*). The P-T probability framework and the G theory have been applied to several areas, such as data compression related to visual discrimination [30], machine learning [31], Bayesian confirmation [33], and the philosophy of science [32].

For overcoming the rate-distortion function’s disadvantages, this paper sets up a transformation relation between the truth function and the distortion function and uses the truth function to replace the distortion function for the constraint condition to obtain the rate-truth function. Since a truth function can also be explained as a membership function, similarity function, confusion probability function, or DCF [32], it can be used as a learning function. It is often expressed as an NEF or Logistic function. In this way, we can overcome rate-distortion functions’ three disadvantages because:the truth function is a learning function; it can be obtained from a sample or sampling distribution [31] and hence is not subjectively defined;using the transformation relation, we can indirectly express the distortion function between any instance *x* and any label *y* by the truth function that may come from machine learning;truth functions indicate labels’ semantic meanings and, hence, can be used as the constraint condition for semantic compression.

Combining the author’s previous studies [31,32], we can explain that: the NEF and the partition functions are the truth function and the logical probability, respectively;the formula, such as Equation (2), for the distribution of Minimum Mutual Information (MMI) or maximum entropy is the semantic Bayes’ formula;MMI *R*(*D*) can be expressed by the semantic mutual information formula;maximum entropy is equal to extremely maximum entropy minus semantic mutual information.

The new explanations are not against the existing explanations but complement them. We can use the fuzzy truth criterion to replace the distortion criterion so that the rate-distortion function becomes the rate-truth function *R*(*Θ*), where *Θ* is a group of fuzzy sets or (fuzzy) truth functions. We can also use the semantic information criterion, which is compatible with the likelihood criterion, to replace the distortion criterion so that the rate-distortion function becomes the rate-verisimilitude function *R*(*G*), where *G* is the lower limit of semantic mutual information. Both functions can be used for semantic compression. The rate-verisimilitude function has been introduced before [30,31], whereas the rate-truth function is provided first in this paper.

This paper mainly aims to:help readers understand rate-distortion functions and maximum entropy distributions from the new perspective;combine machine learning (for the distortion function) and data compression;show that the rate-distortion function can be extended to the rate-truth function and the rate-verisimilitude function for communication data’s semantic compression.

This paper provides an example to show how the Shannon channel matches the semantic channel to achieve MMI *R*(*Θ*) and *R*(*G*) for a given source *P*(*x*) and a group of truth functions. It provides another example to show that the rate-truth function can be used for data reduction (e.g., the compression of decreasing data resolution). The results support the theoretical analyses.

The new explanations should more cohesively combine classical information theory, semantic information G theory, maximum entropy theory, the likelihood method, and fuzzy set theory, with each other. Moreover, the rate-distortion function’s extensions should be practical for semantic compression and helpful for explaining machine learning with NEFs and Soft-max functions. In turn, the new explanations and extensions also support the P-T probability framework and the semantic information G theory.

## 2. Background

### 2.1. Shannon’s Entropies and Mutual Information

**Definition** **1.**

*Variable x denotes an instance; X denotes a discrete random variable taking a value x ∈ U = {x_1_, x_2_, …, x_m_}.*

*Variable y denotes a hypothesis or label; Y denotes a discrete random variable taking a value y ∈ V = {y_1_, y_2_, …, y_n_}.*



Shannon calls *P*(*X*) the source, *P*(*Y*) the destination, *P*(*Y*|*X*) the channel, and *P*(*y_j_*|*x*) (with a certain *y_j_* and variable *x*) a Transition Probability Function (TPF) [34] (p. 18). A Shannon channel is a transition probability matrix or a group of TPFs:*P*(*y|x*): *P*(*y_j_|x*), *j* = 1, 2, …, *n.*

Shannon’s mutual information is:(3)$$\begin{array}{l} I(X;Y)=\sum_{j}\sum_{i}P(x_{i},y_{j})\log\frac{P(x_{i}|y_{j})}{P(x_{i})}=H(X)-H(X|Y)\\ \;\;\;=\sum_{j}\sum_{i}P(x_{i},y_{j})\log\frac{P(y_{j}|x_{i})}{P(y_{j})}=H(Y)-H(Y|X), \end{array}$$
where *H*(*X*) and *H*(*Y*) are Shannon’s entropies of *X* and *Y*; *H*(*X*|*Y*) and *H*(*Y*|*X*) are Shannon’s conditional entropies of *X* and *Y*.

When *Y = y_j_* is given, *I*(*X*; *Y*) becomes the Kullback–Leibler (KL) divergence:(4)$$I(X;y_{j})=\sum_{i}P(x_{i}|y_{j})\log\frac{P(x_{i}|y_{j})}{P(x_{i})}.$$

It is greater or equal to 0. It is equal to 0 as *P*(*x*|*y_j_*) = *P*(*x*).

### 2.2. Rate-Distortion Function R(D)

Shannon proposed the information rate-distortion function [1,2]. Since the rate-distortion function for an i.i.d. source *P*(*X*) and bounded function *d*(*x*, *y*) is equal to the associated information rate-distortion function, Cover and Thomas, in [14] (p. 307), do not distinguish the two functions in most cases. They call both functions the rate-distortion function and use *R*(*D*) to denote them. We follow their example. The following is the definition of the (information) rate-distortion function.

Let the distortion function be *d*(*x*, *y*) or *d_ij_* = *d*(*x_i_*, *y_j_*), *i* = 1, 2, …; *j* = 1, 2, …; $\bar{d}$ be the average of *d*(*x*, *y*); and *D* be the upper limit of $\bar{d}$. An often-used distortion function is *d*(*x*, *y*) = (*x* − *y*)^2^. This function fits cases where *x* and *y* have the same universe (e.g., *U = V*), and distortion only depends on the distance between *x* and *y*.

The MMI for the given *P*(*X*) and *D* is defined as:(5)$$R(D)=\underset{P(y|x):\;\bar{d}\leq D}{\min}I(X;Y)$$

We can obtain the parameter solution of the rate-distortion function by the variational method [2,3]. The constraint conditions are:(6)$$D=\sum_{j}\sum_{i}P(x_{i})P(y_{j}|x_{i})d_{ij}$$
(7)$$\sum_{j}P(y_{j}|x_{i})=1,\;i=1,2,\ldots ,m$$
(8)$$\sum_{j}P(y_{j})=1.$$

The Lagrange function is therefore: (9)$$F=I(X;Y)-sD-\mu_{i}\sum_{j}P(y_{j}|x_{i})-\alpha \sum_{j}P(y_{j}).$$

Since *P*(*y*|*x*) and *P*(*y*) are interdependent, we need to fix one to optimize another. To optimize *P*(*y*|*x*), we fix *P*(*y*) and order $\partial F/\partial P(y_{j}|x_{i})=0$. Then we derive the optimized TPFs or channel:(10)$$\begin{array}{l} P(y_{j}|x_{i})=P(y_{j})\lambda_{i}\exp(sd_{ij}),i=1,2,\ldots ;j=1,2,\ldots \\ \;\;\;\lambda_{i}=1/\sum_{k}P(y_{k})\exp(sd_{ik}), \end{array}$$
where exp( ) is the inverse function of log( ); *λ_i_* is defined with *λ_i_* ≡ exp(*μ_i_*/*P*(*x_i_*)). To optimize *P*(*y*), we fix *P*(*y_j_*|*x_i_*) in *F* and order $\partial F/\partial P(y_{j})=0$. Hence, we derive α = 1 and the optimized *P*(*y*):(11)$$P(y_{j})=\sum_{i}P(x_{j})P(y_{j}|x_{i})$$

Since *P*(*y*|*x*) and *P*(*y*) are interdependent, we first suppose *P*(*y*_1_) = *P*(*y*_2_) = … 1/*n* and then repeat Equations (10) and (11) until *P*(*y*) is unchanged. We call this iteration the MMI iteration.

It is worth noting that our purpose is to find proper TPFs *P*(*y_j_*|*x*), *j* = 1, 2, …, but it is difficult to find them directly because of the constraint conditions in Equations (7) and (8). Therefore, we can only find the proper posterior distributions *P*(*y*|*x_i_*) of *y*, *i* = 1, 2, …, by the MMI iteration to obtain the TPFs indirectly.

The rate-distortion function *R*(*D*) with parameter *s* is [3] (p. 32):(12)$$\begin{array}{l} D(s)=\sum_{i}\sum_{j}d_{ij}P(x_{i})P(y_{j})\exp(sd_{ij})/Z_{i},\\ R(s)=sD(s)-\sum_{i}P(x_{i})\log Z_{i},\\ Z_{i}=1/\lambda_{i}=\sum_{k}P(y_{k})\exp(sd_{ik}). \end{array}$$
where *Z_i_* is the partition function.

The parameter *s* = d*R*/d*D* is negative, and hence exp(*sd_ij_*) is a negative exponential function. Thus, a larger |*s*| will result in narrower exp(*sd_ij_*), larger *R*, and less *D*.

Shannon has proved that *R*(*D*) is the lower limit of the average codeword length for i.i.d. source *P*(*X*) with an average distortion limit, *D*. The rate-distortion theory is the basic theory of data compression for digital communication.

Since *I*(*X*; *Y*) = *H*(*X*) − *H*(*X*|*Y*) and *P*(*X*) is unchanged, minimizing *R* = *I*(*X*; *Y*) is equivalent to maximizing *H*(*X*|*Y*). Therefore, the MMI distribution *P*(*y*|*x*) also maximizes posterior entropy *H*(*X*|*Y*).

### 2.3. The Maximum Entropy Method

Jaynes [4,5] first expounded the maximum entropy method and argued that entropy in statistical mechanics should simply be viewed as a particular application of entropy in information theory.

Supposing that we need to maximize joint entropy *H*(*Y*, *X*) for a given source *P*(*x*), maximizing *H*(*X*, *Y*) is the equivalent to maximizing *H*(*Y*|*X*):(13)$$H(Y|X)=-\sum_{i}\sum_{j}P(x_{i})P(y_{j}|x_{i})\log P(y_{j}|x_{i}).$$

Morevoer, supposing there are feature functions *f_k_*(*x*, *y*), *k* = 1, 2, … the constraint conditions are:(14)$$\sum_{i}\sum_{j}P(x_{i},y_{j})f_{k}(x_{i},y_{j})\geq F_{k},\;k=1,2,\ldots$$
(15)$$\sum_{j}P(y_{j}|x_{i})=1,\;i=1,2,\ldots$$

The Lagrange function is therefore: (16)$$F=H(Y|X)-\sum_{k}\alpha_{k}\sum_{i}\sum_{j}P(x_{i})P(y_{j}|x_{i})f_{k}(x_{i},y_{j})-\mu_{i}\sum_{j}P(y_{j}|x_{i})$$

By ordering $\partial F/\partial P(y|x_{i})=0$, we derive the optimized channel *P*(*y*|*x*):(17)$$\begin{array}{l} P(y|x_{i})=\exp[\sum_{k}\alpha_{k}f_{k}(x_{i},y)]/Z_{i},i=1,2,\ldots \\ {\;\;\;Z}_{i}=\sum_{k}\exp(\sum_{j}\alpha_{k}f_{k}(x_{i},y_{j}). \end{array}$$

This *P*(*y*|*x*) can maximize *H*(*X*, *Y*).

## 3. The Author’s Related Work

### 3.1. The P-T Probability Framework

The semantic information G theory is based on the P-T probability framework [29,30]. This framework includes two types of probabilities: the statistical probability denoted by *P* and the logical probability by *T*.

**Definition** **2.**
*(for the P-T probability framework):*
*The y_j_ is a label or a hypothesis; y_j_(x_i_) is a proposition. The θ_j_ is a fuzzy subset of universe U, whose elements make y_j_ true. We have y**_j_(x) ≡**“x**∈ θ_j_” ≡ “x belongs to θ_j_”(“≡**” means they are* logically equivalent *according to the definition).*
*The θ_j_ may also be a model or a group of model parameters.*
*A probability that is defined with “=”, such as P(y_j_) ≡ P(Y = y_j_), is a statistical probability. A probability that is defined with “*
*∈”, such as P(X ∈ θ_j_), is a logical probability. To distinguish P(Y = y_j_) and P(X ∈ θ_j_), we define T(y_j_) ≡ T(θ_j_) ≡ P(X ∈ θ_j_) as the logical probability of y_j_.*

*T(y_j_|x) ≡ T(θ_j_|x) ≡ P(X ∈ θ_j_|X = x) is the truth function of y_j_ and the membership function of θ_j_. It changes between 0 and 1, and the maximum of T(y|x) is 1.*



A semantic channel consists of a group of truth functions:*T*(*y*|*x*): *T*(*θ_j_*|*x*), *j* = 1, 2, …, *n*.

According to the above definition, we have the logical probability:(18)$$T(y_{j})\equiv T(\theta_{j})\equiv P(X\in \theta_{j})=\sum_{i}P(x_{i})T(\theta_{j}|x_{i})$$

Zadeh calls this probability the fuzzy event’s probability [35]. If *θ_j_* is a crispy set, *T*(*θ_j_*) becomes the cumulative probability or two cumulative probabilities’ difference [36].

Generally, *T*(*y*_1_) + *T*(*y*_2_) + … + *T*(*y_n_*) > 1. For example, the sum of the logical probabilities of four labels in Figure 1 is greater than 1 since *T*(*y*_1_) + *T*(*y*_3_) = 1. The detailed discussions about the distinctions and relations between statistical probability and logical probability can be found in [32].

We can put *T*(*θ_j_*|*x*) and *P*(*x*) into the Bayes’ formula to obtain a likelihood function:(19)$$P(x|\theta_{j})=\frac{T(\theta_{j}|x)P(x)}{T(\theta_{j})},\;T(\theta_{j})=\sum_{i}T(\theta_{j}|x_{i})P(x_{i})$$

*P*(*x*|*θ_j_*) is called the semantic Bayes’ prediction. It is often written as *P*(*x*|*y_j_*, *θ*) in popular methods. We call the above formula the semantic Bayes’ formula. 

Since the maximum of *T*(*y*|*x*) is 1, from *P*(*x*) and *P*(*x*|*θ_j_*), we can obtain:(20)$$T(\theta_{j}|x)=\frac{T(\theta_{j})P(x|\theta_{j})}{P(x)},\;T(\theta_{j})=1/\max[P(x|\theta )/P(x)],$$
where max[*P*(*x*|*θ*)/*P*(*x*)] means the maximum of *P*(*x*|*θ*)/*P*(*x*) for different *x* and *y*. In the author’s earlier articles [31], *T*(*θ_j_*) = 1/max[*P*(*x*|*θ_j_*)/*P*(*x*)]. This change in Equation (20) can ensure that truth functions are symmetrical, e.g., *T*(*x*|*y*) = *T*(*y*|*x*), as well as distortion functions. This change is also for comparing two truth functions *T*(*y_j_*|*x*) and *T*(*y_k_*|*x*) for classification according to the correlation between *x* and *y* (see Equation (23)). Since *P*(*x*|*θ_j_*) in Equation (19) is changeless, as *T*(*θ_j_*|*x*) is replaced with *cT*(*θ_j_*|*x*) (where *c* is a positive constant), this change does not influence the other uses of *T*(*θ_j_*|*x*).

Equations (19) and (20) form the third Bayes’ theorem [31], can be used to convert the likelihood function and the truth function from one to another.

### 3.2. The Semantic Information G Measure

The author [28] defines the (amount of) semantic information conveyed by *y_j_* in relation to *x_i_* with log-normalized-likelihood: (21)$$I(x_{i};\theta_{j})=\log\frac{P(x_{i}|\theta_{j})}{P(x_{i})}=\log\frac{T(\theta_{j}|x_{i})}{T(\theta_{j})}$$

The value *I*(*x_i_*; *θ_j_*), or its average, is the semantic information G measure or, simply, the G measure. If *T(θ_j_*|*x*) is always 1, the G measure becomes Carnap and Bar-Hillel’s semantic information measure [21].

The above formula is illustrated in Figure 2. Figure 2 indicates that the less the logical probability is (e.g., the lower the horizontal line is), the more information there is available; the larger the deviation is, the less information there is available; a wrong hypothesis conveys negative information. These conclusions accord with Popper’s thoughts [37] (p. 294). For this reason, *I*(*x_i_*; *θ_j_*) is also explained as the verisimilitude between *y_j_* and *x_i_* [32].

We can also use the above formula to measure sensory information, for which *T*(*θ_j_*|*x*) is the confusion probability function of *x_j_* with *x* or the discrimination function of *x_j_* [31].

By averaging *I*(*x_i_*; *θ_j_*), we obtain generalized KL information:(22)$$I(X;\theta_{j})=\sum_{i}P(x_{i}|y_{j})\log\frac{P(x_{i}|\theta_{j})}{P(x_{i})}=\sum_{i}P(x_{i}|y_{j})\log\frac{T(\theta_{j}|x_{i})}{T(\theta_{j})}$$
where *P*(*x_i_*|*y_j_*), *i* = 1, 2, …, is the sampling distribution, which may be unsmooth or discontinuous. It is easy to prove *I*(*X*; *θ_j_*) ≤ *I*(*X*; *y_j_*) [31].

When the sample is enormous, so that *P*(*x*|*y_j_*) is smooth, we may let *P*(*x|θ_j_*) = *P*(*x|y_j_*) or *T*(*θ_j_*|*x*) ∝ *P*(*y_j_*|*x*) to obtain the optimized truth function:(23)$$\begin{array}{l} T*(\theta_{j}|x)=\frac{P^{*}(x|\theta_{j})}{P(x)}/\max\left(\frac{P^{*}(x|\theta )}{P(x)}\right)\\ \;\;\;\;=\frac{P(x|y_{j})}{P(x)}/\max\left(\frac{P(x|y)}{P(x)}\right)=\frac{P(x,y_{j})}{P(x)P(y_{j})}/\max\left(\frac{P(x,y)}{P(x)P(y)}\right). \end{array}$$

According to this equation, *T**(*θ_j_*|*x_i_*) = *T**(*θ_xi_*|*y_j_*) or *T**(*y_j_*|*x_i_*) = *T**(*x_i_*|*y_j_*), where *θ_xi_* is a fuzzy subset of *V*. Furthermore, we have: *T**(*θ_j_*|*x*) = *cP*(*y_j_*|*x*)(24)
where *c* is a constant. The above formula means the optimized truth function *T**(*θ_j_|x*) is proportional to TPF *P*(*y_j_*|*x*). This formula accords with Wittgenstein’s thoughts: meaning lies in uses [38] (p. 80).

If *P*(*x*|*y_j_*) is unsmooth, we may achieve a smooth *T**(*θ_j_*|*x*) with parameters by:(25)$$T*(\theta_{j}|x)=\underset{T(\theta_{j}|x)}{\arg\max}\sum_{i}P(x_{i}|y_{j})\log\frac{T(\theta_{j}|x_{i})}{T(\theta_{j})}.$$

By averaging *I*(*X*; *θ_j_*) for different *y*, we obtain semantic mutual information:(26)$$\begin{array}{l} I(X;Y_{\theta })=\sum_{j}P(y_{j})\sum_{i}P(x_{i}|y_{j})\log\frac{P(x_{i}|\theta_{j})}{P(x_{i})}\\ \;\;\;=\sum_{i}\sum_{j}P(x_{i})P(y_{j}|x_{i})\log\frac{T(\theta_{j}|x_{i})}{T(\theta_{j})}=H(Y_{\theta })-H(Y_{\theta }|X), \end{array}$$
where: (27)$$H(Y_{\theta })=-\sum_{j}P(y_{j})\log T(\theta_{j}),$$
(28)$$H(Y_{\theta }|X)=-\sum_{j}\sum_{i}P(x_{i},y_{j})\log T(\theta_{j}|x_{i})$$

*H*(*Y_θ_*) is a cross-entropy. Since ∑*_j_ T*(*θ_j_*) ≥ 1, we also call *H*(*Y_θ_*) a generalized entropy or a semantic entropy. *H*(*Y**_θ_*|*X*) is called a fuzzy entropy.

When we fix the Shannon channel *P*(*y*|*x*) and let *P*(*x*|*θ_j_*) = *P*(*x*|*y**_j_*) or *T*(*θ_j_*|*x*) ∝ *P*(*y_j_*|*x*) for every *j* (Matching I), *I*(*X*; *Y_θ_*) reaches its maximum *I*(*X*; *Y*). If we use a group of truth functions or a semantic channel *T*(*y*|*x*) as the constraint function to seek MMI, we need to let *P*(*x*|*y**_j_*) = *P*(*x*|*θ_j_*) or *P*(*y_j_*|*x*) ∝ *T*(*θ_j_*|*x*) as far as possible for every *j* (Matching II). Section 4.3 and Section 4.4 further discuss Matching II.

Letting *T*(*θ_j_*|*x*) = exp[−(*x* − *x_j_*)^2^/(2*σ*^2^)], we have:(29)$$\begin{array}{l} I(X;Y_{\theta })=H(Y_{\theta })-H(Y_{\theta }|X)\\ \;\;\;=-\sum_{j}P(y_{j})\log T(\theta_{j})-\sum_{j}\sum_{i}P(x_{i},y_{j}){\left(x_{i}-\mu_{j}\right)}^{2}/(2\sigma_{j}{}^{2}). \end{array}$$

It is easy to find that the above mutual information is like the Regularized Least Square (RLS). *H*(*Y_θ_*) is like the regularization term, and the next one is the relative squared error term. Therefore, we can treat the maximum semantic mutual information criterion as a particular RLS criterion.

## 4. Theoretical Results

### 4.1. The New Explanations of the MMI Distribution and the Rate-Distortion Function

It is easy to regard exp[*sd*(*x_i_*, *y*)] in the rate-distortion function as a truth function and *Z_i_* as a logical probability. We can let *θ**_xi_* be a fuzzy subset of *V* (*V* = {*y*_1_, *y*_2_, …}) and *T*(*x_i_*|*y*) ≡ *T*(*θ_xi_*|*y*) be the truth function of *y*(*x_i_*), and *T(x_i_*) = *T(θ_xi_*) = ∑*_j_ P(y_j_*)*T(y_j_*|*x_i_*) be the logical probability of *x_i_*. Hence, we can observe that the MMI distribution *P*(*y*|*x*) in the rate-distortion function is produced by the semantic Bayes’ formula:*P*(*y*|*x_i_*) = *P*(*y*)exp[*sd*(*x_i_*, *y*)]/*Z_i_* = *P*(*y*)*T*(*x_i_*|*y*)/*T*(*x_i_*), *i* = 1, 2, …, *m*.(30)

*R*(*D*) can be expressed by the semantic mutual information formula because:(31)$$\begin{array}{l} I(Y;X_{\theta })=\sum_{j}\sum_{i}P(x_{i},y_{j})\log[T(x_{i}|y_{j})/T(x_{i})]\\ \;\;\;=\sum_{j}\sum_{i}P(x_{i},y_{j})\log[\exp(sd(x_{i},y_{j})]-\sum_{i}P(x_{i})\ln Z_{i}\\ \;\;\;=sD(s)-\sum_{i}P(x_{i})\log Z_{i}=R(D). \end{array}$$

### 4.2. Setting Up the Relation between the Truth Function and the Distortion Function

We can now improve the G theory by setting up the relation between the truth function and the distortion function.

Four truth functions in Figure 1 also reveal the distortion when we use *y_j_* to represent *x_i_*. If the truth value of *y_j_*(*x_i_*) is 1, the distortion *d(x_i_*, *y_j_*) should be 0. The distortion increases as the truth value decreases. Hence, we use the following definition:

**Definition** **3.**
*The transformation relation between the distortion function and the truth function is defined as:*

*d*(*x*, *y*) ≡ log[1*/T*(*y*|*x*)].(32)


According to this definition, we have *T*(*y*|*x*) = exp[−*d*(*x*, *y*)] and *H*(*Y_θ_*|*X*) = $\bar{d}$. Therefore, we can use *H*(*Y_θ_*|*X*) to replace $\bar{d}$ for the constraint condition.

Since *T(y*|*x*) = *T(x*|*y*), we also have *d*(*x*, *y*) = log[1/*T*(*x*|*y*)].

### 4.3. Rate-Truth Function R(Θ)

The author has proposed the rate-of-limiting-error function previously in [30], an immature study. This function is more akin to the extension of the complexity-distortion function [39], unlike the rate-truth function, which is the extension of the rate-distortion function.

In the following, we use almost the same method used for *R*(*D*) to obtain the rate-truth function *R*(*Θ*), where *Θ* means a group of truth functions or fuzzy sets. The constraint condition $\bar{d}\leq D$ becomes:(33)$$H(Y_{\Theta }|X)=-\sum_{i}\sum_{j}P(x_{i})P(y_{j},x_{i})\log T(y_{j}|x_{i})\leq D$$

Following the rate-distortion function’s derivation (see Equation (10)), we obtain the optimized posterior distribution *P*(*y*|*x_i_*) of *y*:(34)$$\begin{array}{l} P(y|x_{i})=P(y)T{\left(y|x_{i}\right)}^{\left|s\right|}/\sum_{k}P(y_{k})T{\left(y_{k}|x_{i}\right)}^{\left|s\right|}\\ \;\;\;\;\;\;\;\;\;\;\;\;\;\;\;\;\;i=1,2,\ldots \\ \;\;\;=P(y)T{\left(x_{i}|y\right)}^{\left|s\right|}/T(x_{i})=P(y|\theta_{xi}), \end{array}$$
where we replace −*s* with |*s*|, which is positive. The larger the |*s*| value is, the clearer the boundaries of the fuzzy sets are, and hence the larger the *R* is.

Next, we obtain the optimized *P*(*y*) (see Equation (11)).

We can obtain the Shannon channel *P*(*y*|*x*) that minimizes *R* by repeating Equations (11) and (34) until *P*(*y*) converges, e.g., by the MMI iteration.

Bringing *P*(*y*|*x_i_*) in Equation (34) into the mutual information formula, we obtain the rate-truth function *R*(*Θ*):(35)$$\begin{array}{l} R(\Theta )=\sum_{i}\sum_{j}P(x_{i})P(y_{j}|x_{i})\log[P(y_{j}|x_{i})/P(y_{j})]\\ \;\;=\sum_{i}\sum_{j}P(x_{i})P(y_{j}|x_{i})\log[T{\left(x_{i}|y_{j}\right)}^{\left|s\right|}/T(x_{i})]\\ \;\;=H(X_{\theta })-H(X_{\theta }|Y)=I(Y;X_{\theta }), \end{array}$$
where:(36)$$\begin{array}{l} H(X_{\theta }|Y)=-\sum_{i}\sum_{j}P(x_{i},y_{j})\log T{\left(x_{i}|y_{j}\right)}^{\left|s\right|}\\ T(x_{i})=\sum_{j}T{\left(x_{i}|y_{j}\right)}^{\left|s\right|}P(y_{j}),\\ H(X_{\theta })=-\sum_{i}P(x_{i})\log T(x_{i}). \end{array}$$

We have *R*(*Θ*) = *H*(*X_θ_*) − *H*(*X_θ_*|*Y*) = *I*(*Y*; *X_θ_*), which means that the MMI can be expressed by semantic mutual information: *I*(*Y*; *X_θ_*). The constraint is tighter when |*s*| > 1 and looser when |*s*| < 1, than the constraint |*s*| = 1. When the constraint condition is the original truth function without *s* or with *s* = −1, we have:*R*(*Θ*) = *I*(*X*; *Y*) = *I*(*Y*; *X_θ_*) ≥ *I*(*X*; *Y_θ_*).(37)

The reason for *I*(*Y*; *X_θ_*) ≥ *I*(*X*; *Y_θ_*) is that the iteration can only let *P*(*y_j_*|*x*) be proximately proportional to *T*(*y_j_*|*x*), and not exactly in general (see Equation (34) and Section 5.2).

According to Shannon’s lossy coding theorem of discrete memoryless sources [14] (p. 307), for the given sources *P*(*X*) and *D*, we can use block coding to achieve a minimum average codeword length, whose lower limit is *R*(*D*). Since *H*(*X_θ_*|*Y*) can be understood as the average distortion, *R*(*Θ*) also means the lower limit of average codeword length.

For any distortion function *d*(*x*, *y*), we can always express the corresponding truth function as exp[−*d*(*x*, *y*)]. But for truth functions, such as those in Figure 1, we may not be able to express them by a distortion function. Therefore, the rate-distortion function may be regarded as the rate-truth function’s particular case, as the truth function is exp[−*d*(*x*, *y*)].

### 4.4. Rate-Verisimilitude Function R(G)

If we change the average distortion criterion into the semantic mutual information criterion, which is compatible with the likelihood criterion, then the rate-distortion function becomes the rate-verisimilitude function [31]. In this case, we replace *d_ij_* = *d*(*x_i_*, *y_j_*) with *I_ij_* = *I*(*x_i_*; *θ_j_*). The constraint condition $\bar{d}\leq D$ becomes *I*(*X*; *Y_ϴ_*) ≥ *G*, where *G* denotes the lower limit of semantic mutual information. Following the rate-distortion function’s derivation, we can obtain:(38)$$\begin{array}{l} G(s)=\sum_{i}\sum_{j}P(x_{i})P(y_{j}|x_{i})I_{ij}=\sum_{i}\sum_{j}I_{ij}P(x_{i})P(y_{j})\exp(sI_{ij})/Z_{i},\\ R(s)=sG(s)-\sum_{i}P(x_{i})\log Z_{i}, \end{array}$$
where *s* is positive, and:(39)$$\begin{array}{l} P(y_{j}|x_{i})=P(y_{j}){\left[\frac{T(y_{j}|x_{i})}{T(y_{j})}\right]}^{s}/Z_{i},\;i=1,2,\ldots ;j=1,2,\ldots \\ \;\;{\;Z}_{i}=\sum_{k}P(y_{k}){\left[\frac{T(y_{k}|x_{i})}{T(y_{k})}\right]}^{s}. \end{array}$$

We also need the MMI iteration to optimize *P*(*y*) and *P*(*y*|*x*). The function *P*(*y*|*x_i_*) is now like a Softmax function, in which the numerator *P*(*y*)[*T*(*y*|*x_i_*)/*T*(*y*)]^s^ may be greater than 1. In the E-step of the EM algorithm for mixture models, there is a similar formula, which is also used for MMI [40].

*R*(*G*) is more suitable than *R*(*D*) and *R*(*Θ*) when *y* is a prediction, such as a weather prediction, where information is more important than truth. More discussions and applications of the *R*(*G*) function can be found in [30,31].

### 4.5. The New Explanation of the Maximum Entropy Distribution and the Extension

We consider maximizing joint entropy *H*(*X*, *Y*) for the given *P*(*x*).

Consider Equation (17) for the maximum entropy distribution. We may assume that *P*(*y*) is a constant (1/*n*) so that *H*(*Y*) is the maximum. Then Equation (17) becomes:(40)$$\begin{array}{l} P(y|x_{i})=P(y)\exp(\sum_{k}\alpha_{k}f_{k}(x_{i},y))/Z_{i}^{\prime },i=1,2,\ldots \\ \;\;Z_{i}^{\prime }=\sum_{k}P(y)\exp(\sum_{j}\alpha_{k}f_{k}(x_{i},y_{j})). \end{array}$$

The above exp(.) is an NEF. We can explain exp(.) as a truth function, *Z_i_*′ as a logical probability, and the above formula as a semantic Bayes’ formula.

Then, maximum joint entropy can be expressed as:(41)$$\begin{array}{l} H(X,Y)=H(X)+H(Y|X)=H(X)-\sum_{i}P(x_{i})\sum_{j}P(y_{j}|x_{i})\log P(y_{j}|x_{i})\\ =H(X)-\sum_{i}\sum_{j}P(x_{i},y_{i})\log[P(y_{j})\exp(\sum_{j}\alpha_{k}f_{k}(x_{i},y_{j}))/Z_{i}^{\prime }]\\ =H(X)+H(Y)-I(Y;X_{\theta }), \end{array}$$
where *H*(*Y*) is equal to log *n*, *I*(*X*; *Y_θ_*) is semantic mutual information, and *H*(*X*) + *H*(*Y*) is the extremely maximum value of *H*(*X*, *Y*). Therefore, we can explain that maximum entropy is equal to extremely maximum entropy minus semantic mutual information.

If we use a group of truth functions or membership functions, such as those in Figure 1, as the constraint condition, to extend the maximum entropy method, the above distribution becomes:(42)$$P(y|x_{i})=T{\left(y|x_{i}\right)}^{\left|s\right|}/\sum_{k}T{\left(y_{k}|x_{i}\right)}^{\left|s\right|},\;i=1,\;2,\;\ldots \;$$

This distribution is the same as in Equation (34) with *P*(*y*) = 1/*n*. 

### 4.6. The New Explanation of the Boltzmann Distribution and the Maximum Entropy Law

Using Stirling’s formula ln*N*! = *N*ln*N* − *N* as *N* → ∞, Jaynes proved that Boltzmann’s entropy and Shannon’s entropy have a simple relation [4,5]:(43)$$S=k\ln W=k\ln\frac{N!}{\prod_{i}N_{i}!}=-kN\sum_{i}P(x_{i}|T)\ln P(x_{i}|T)=kNH(X|T),$$
where *k* is the Boltzmann constant, *W* is the microstate number, *x_i_* means state *i*, *N* is the particles’ number, and *P*(*x_i_*|*T*) denotes the probability of a particle (or the density of particles) in state *i* for the given absolute temperature *T*.

The Boltzmann distribution [41] is:(44)$$P(x_{i}|T)=\exp(-\frac{e_{i}}{kT})/Z,Z=\sum_{i}\exp(-\frac{e_{i}}{kT}),$$
where *Z* is the partition function.

If *x_i_* means energy *e_i_*, *G_i_* is the number of states with *e_i_*, and *G* is the total number of all states, then *P*(*x_i_*) = *G_i_*/*G* is the prior probability of *x_i_*. Hence, Equations (43) and (44) become:(45)$$\begin{array}{l} S=k\ln\frac{N!}{\prod_{i}N_{i}!/G_{i}{}^{N_{i}}}=-kN\sum_{i}P(x_{i}|T)\ln\frac{P(x_{i}|T)}{G_{i}}\\ \;=-kN\sum_{i}P(x_{i}|T)\ln\frac{P(x_{i}|T)}{P(x_{i})}+kN\ln G, \end{array}$$
(46)$$P(x_{i}|T)=P(x_{i})\exp(-\frac{e_{i}}{kT})/Z^{\prime },Z^{\prime }=\sum_{i}P(x_{i})\exp(-\frac{e_{i}}{kT})$$

Now, we can explain exp[−*e_i_*/(*kT*)] as a truth function *T*(*θ_j_*|*x*), *Z*′ as a logical probability *T*(*θ_j_*), and Equation (46) as a semantic Bayes’ formula.

Assuming that for a local equilibrium system, its different areas *y_j_*, *j* = 1, 2, …, have different temperatures *T_j_*, *j* = 1, 2, … The above *G* becomes *G_j_*. Then we can derive (see Appendix B for the details):(47)$$S/(kN)=\sum_{j}P(y_{j})\ln G_{j}-I(X;Y_{\theta })$$
which means that the thermodynamic entropy *S* is proportional to the extremely maximum entropy $\sum_{j}P(y_{j})\ln G_{j}$ minus the semantic mutual information *I*(*X*; *Y_θ_*). This formula indicates that the maximum entropy law in physics can be equivalently stated as the MMI law; this MMI can be expressed as semantic mutual information *I*(*X*; *Y_θ_*).

## 5. Experimental Results

### 5.1. An Example Shows the Shannon Channel’s Changes in the MMI Iterations for R(Θ) and R(G)

The author experimented with Example 1 to test two theoretical results:the MMI iteration lets the Shannon channel match the semantic channel, e.g., lets *P*(*y_j_*|*x*) ∝ *T*(*y_j_*|*x*), *j* = 1, 2, …, as far as possible for the functions *R*(*D*), *R*(*Θ*), and *R*(*G*);the MMI iteration can reduce mutual information.

**Example** **1.***The four truth functions* and the population age distribution P(x) are *shown in*
Figure 1
*(see*
Appendix C
*for the formulas producing these lines).*
*The task is to use the above truth functions as the constraint condition to obtain P(y|x) for R(Θ) and P(y|x) for R(G).*

First, the author uses the above truth functions as the constraint condition for the *R*(*Θ*) with |*s*| = 1. The iteration process for *P*(*y*|*x*) is shown in Figure 3 (see Appendix A to find how the data for Figure 3 are produced). 

The convergent *P*(*y*) is {*P*(*y*_1_), *P*(*y*_2_), *P*(*y*_3_), *P*(*y*_4_)} = {0.3499, 0.0022, 0.6367, 0}. The MMI is 0.845 bits. The iterative process not only lets every *P*(*y_j_*|*x*) be approximatively proportional to *T*(*y_j_*|*x*) but also makes *P*(*y*_4_) closer to 0. The value of *P*(*y*_4_) became 0 because *y*_4_ implies *y*_3_ and hence can be replaced with *y*_3_. The latter has a larger logical probability. By replacing *y*_4_ with *y*_3_, we can save the average codeword length.

Figure 3c indicates that *H*(*Y*) also decreases with the iteration. Since *H*(*Y*) is much less than *P*(*X*), we can also simply replace the instance *x* with the label *y* to compress data.

To maximize the entropy *H*(*X*, *Y*) or *H*(*Y*|*X*) for a given *P*(*x*), we do not need the iteration for *P*(*y*|*x*). The function *P*(*y*|*x*) in Figure 3a also results in the maximum entropies *H*(*Y*|*X*) and *H*(*X*, *Y*), where *P*(*y*|*x*) matches *T*(*y*|*x*) only once.

Next, the author uses the above truth functions as the constant condition for the *R*(*G*) with |*s*| = 1. For *R*(*G*), *d*(*x_i_*, *y_j_*) is replaced with *I*(*x_i_*; *θ_j_*) = log[*T*(*y_j_*|*x_i_*)/*T*(*y_j_*)] for every *i* and *j*. Figure 4 shows the iterative results and the process (see Appendix A to find how the data for Figure 4 are produced).

Figure 4a intuitively displays that *P*(*y*|*x*) accords with the four labels’ semantic meanings. The convergent distribution *P*(*y*) for *R*(*G*) is {*P*(*y*_1_), *P*(*y*_2_), *P*(*y*_3_), *P*(*y*_4_)} = {0.3619, 0.0200, 0.6120, 0.0057}. The MMI is 0.883 bits. This *P*(*y*_4_) is not 0. Both *P*(*y*_2_) and *P*(*y*_4_) are larger than those for *R*(*Θ*). The reason is that a label with less logical probability can convey more semantic information and, hence, should be more frequently selected if we use the semantic information criterion. For the same *s* = 1 and *Θ*, *R*(*G*) = 0.883 bits is greater than *R*(*Θ*) = 0.845 bits.

### 5.2. An Example about Greyscale Image Compression Shows the Convergent P(y|x) for the R(Θ)

This example was used to explain how we replace the distortion function with the truth function to obtain *R*(*Θ*) and the corresponding channel *P*(*y*|*x*) for the image compression of decreasing the resolution of pixels’ grey levels. The conclusion is also suitable for data reduction, where too high a resolution is unnecessary.

**Example** **2.***The goal is to compress an 8-bit greyscale image with 256 grey levels (denoted by x_i_, i = 0, 1, …, 255) into a 3-bit grey image with 8 grey levels (denoted by y_j_, j = 1, 2, …, 8)* [42]. *Considering that human visual discrimination changes with grey levels (the higher the grey level is, the lower the discrimination is). We use 8 truth functions, as shown in* Figure 5a, *to represent 8 fuzzy classes.* Appendix D *shows how these lines are produced. The task is to solve the MMI R and the corresponding channel P(y|x) with s = 1.*

The author obtains the convergent *P*(*y*|*x*), as shown in Figure 5b. Figure 5c shows that *R*(*Θ*), *I*(*X*; *Y**_ϴ_*), and *H*(*Y*) change in the iterative process (see Appendix A to find how the data for Figure 5 are produced).

Figure 5 shows how the Shannon channel matches the semantic channel for MMI. Comparing Figure 5a,b, we find that it is easy to control *P*(*y*|*x*) with *T*(*y*|*x*). If we use the distortion function *d*(*x*, *y_j_*) instead of the truth function *T*(*y_j_*|*x*), *j* = 1, 2, …, 8, it is not easy to design *d*(*x*, *y_j_*). It is also difficult to predict the convergent *P*(*y*|*x*) according to *d*(*x*, *y*).

In this example, *I*(*X*; *Y*) = *I*(*Y*; *X_θ_*) is difficult to approach *I*(*X*; *Y_θ_*) because *P*(*y_j_*|*x*) cannot be strictly proportional to *T*(*y_j_*|*x*) for *j* = 2, 3, …, 6.

The author has used different prior distributions and semantic channels to calculate Shannon channels for MMI using the above method and achieving similar results. Every convergent TPF *P*(*y_j_*|*x*) covers an area in *U* identical to the area covered by *T*(*y_j_*|*x*) as *s* = 1, and hence, *P*(*y*|*x*) is satisfied with the constraint condition defined by *T*(*y*|*x*).

## 6. Discussions

### 6.1. Connecting Machine Learning and Data Compression by Truth Functions

Researchers have applied the rate-distortion theory to machine learning and achieved meaningful results [43,44,45]. However, we also need to obtain the distortion function from machine learning.

In the rate-distortion theory, the distortion function *d*(*x*, *y*) is subjectively defined, lacking the objective standard. When the instance *x* and label *y* have different universes, *U* and *V* and |*U*| >> |*V*|, the definition of the distortion between *x* and *y* is problematic. Now we can obtain labels’ truth functions from sampling distributions by machine learning (see Equations (23)–(25)) and indirectly express distortion functions using truth functions (see Equation (32)). After the source *P*(*x*) is changed, the semantic channel *T*(*y*|*x*) is still functional [31]. We can use *T*(*y*|*x*) and the new *P*(*x*) to obtain the new Shannon channel *P*(*y*|*x*) for MMI *R*(*Θ*). We can also use *T*(*y*|*x*)^|*s*|^ with |*s*| > 1 to reduce the average distortion or with |*s*| < 1 to loosen the distortion limit. In this way, we can overcome the rate-distortion function’s three disadvantages mentioned in Section 1. In addition, a group of truth functions describe a group of labels’ extensions and semantic meanings and are therefore intuitional. They have stronger descriptive power than distortion functions because we can always use a truth function to replace a distortion function by *T*(*y_j_*|*x*) = exp[−*d*(*x*, *y_j_*)]. Although, we may not be able to use a distortion function to replace a truth function, such as the truth function of *y*_3_ = “Adult” in Figure 1.

Two often-used learning functions are the truth function (or the similarity function, or the membership function) and the likelihood function. We can also replace the distortion function *d*(*x*, *y*) with a log-likelihood function log*P*(*x*|*θ*), or equivalently replace $\bar{d}$ ≤ *D* with *I*(*X*; *Y_θ_*) ≥ *G* to obtain the rate-verisimilitude function *R*(*G*). The function *R*(*G*) uses the likelihood criterion or the semantic information criterion. It can reduce the underreports of events with less logical probabilities. For example, the selected probabilities *P*(*y*_2_) and *P*(*y_4_*) in Figure 1 for *R*(*G*) (see Figure 4a) are higher than *P*(*y*_2_) and *P*(*y_4_*) for *R*(*Θ*) (see Figure 3c). The function *R*(*G*) is more suitable than *R*(*D*) and *R*(*Θ*) when *y* is a prediction where information is more important than truth.

Like *R*(*D*), *R*(*Θ*) and *R*(*G*) also mean the lower limit of average codeword length.

### 6.2. Viewing Rate-Distortion Functions and Maximum Entropy Distributions from the Perspective of Semantic Information G Theory

There are many similarities between the MMI distribution in the rate-distortion function and the maximum entropy distribution. According to the analyses in Section 4.1 and Section 4.5, we can regard NEFs as truth functions and partition functions as logical probabilities. These distributions or Shannon’s channels, such as *P*(*y*|*x*) in Equation (2), are semantic Bayes’ distributions produced by the semantic Bayes’ formula (see Equation (19)).

The main differences between the rate-distortion theory and the maximum entropy method are:For the rate-distortion function *R*(*D*), we seek MMI *I*(*X*; *Y*) = *H*(*X*) − *H*(*X*|*Y*), which is equivalent to maximizing the posterior entropy *H*(*X*|*Y*) of *X.* For *R*(*D*), we use an iterative algorithm to find the proper *P*(*y*). However, in the maximum entropy method, we maximize *H*(*X*, *Y*) or *H*(*Y*|*X*) for given *P*(*x*) without the iteration for *P*(*y*);The rate-distortion function can be expressed by the semantic information G measure (see Equation (31)). In contrast, the maximum entropy is equal or proportional to the extremely maximum entropy minus semantic mutual information (see Equations (41) and (47)).

### 6.3. How the Experimental Results Support the Explanation for the MMI Iterations

The theoretical analyses in Section 4.2, Section 4.3 and Section 4.4 indicate that the MMI iterations for *R*(*D*), *R*(*Θ*), and *R*(*G*) impel the Shannon channel to match the semantic channel, e.g., impel *P*(*y_j_*|*x*) ∝ *T*(*y_j_*|*x*), *j* = 1, 2, … as far as possible. Example 1 in Section 5.1 shows that the iterative processes for *R*(*Θ*) and *R*(*G*) not only find proper probability distribution *P*(*y*) (which means a label *y_j_* with larger logical probability will be selected more frequently) but also modify every TPF *P*(*y_j_*|*x*) so that *P*(*y_j_*|*x*) is approximatively proportional to *T*(*y_j_*|*x*). Therefore, the results in Figure 3 support the above theoretical analyses.

Figure 4 indicates that the Shannon channel *P*(*y*|*x*) for *R*(*G*) is a little different from that for *R*(*Θ*). For example, two labels *y*_2_ and *y*_4_, with less logical probabilities, have larger *P*(*y*_2_) and *P*(*y*_4_) than those for *R*(*Θ*). This result also accords with the theoretical analysis, which indicates that the semantic information criterion can reduce the underreports of events with less logical probabilities.

Figure 5 shows an example of data reduction. For this example, it is hard to define distortion function *d*(*x*, *y*), but it is easy to use truth functions to represent fuzzy classes. The results indicate that we can easily control the convergent Shannon channel *P*(*y*|*x*) with the semantic channel *T*(*y*|*x*).

### 6.4. Considering Other Kinds of Semantic Information

Wyner and Debowski have independently defined an information measure (see Equation (1) in [27]), which is meaningful for measuring the semantic information between two sentences or labels. Combining this with the G theory, we may develop a different measure from the above for the semantic information between two labels *y_j_* and *y_k_*:(48)$$I(\theta_{j};\theta_{k})=\sum_{i}P(x_{i}|y_{j},y_{k})\log\frac{T(\theta_{j}\cap \theta_{k}|x_{i})}{T(\theta_{j})T(\theta_{k})}.$$

Unlike Equation (1) in [27], the above formula includes both the statistical probabilities, (*P*(*x*) and *P*(*x*|*y_j_*, *y_k_*)), and logical probabilities, and it does not require that subsets *θ*_1_, *θ*_2_, …, *θ_n_* form a partition of *U*. We use an example to explain the reason for using *P(x*). If *P*(*x*) is a population age distribution, we ca use two labels *y*_1_ = “Person in his 50 s” and *y*_2_ = “Elder person” for a person at age *x*. The average life span (denoted by *t*) in different areas and eras might change from 50 years to 80 years. The above formula can ensure that the semantic information between *y*_1_ and *y*_2_ for *t* = 50 is more than that for *t* = 80.

Averaging *I(θ_j_*; *θ_k_*), we have semantic mutual information:(49)$$I(Y_{\theta 1};Y_{\theta 2})=\sum_{j}\sum_{k}\sum_{i}P(x_{i},y_{j},y_{k})\log\frac{T(\theta_{j}\cap \theta_{k}|x_{i})}{T(\theta_{j})T(\theta_{k})}.$$

This formula can ensure that wrong translations or replacements may convey negative semantic information.

The distortion function between the two labels becomes:(50)$$d(y_{j},y_{k})=\sum_{i}P(x_{i}|y_{j},y_{k})\log[1/T(\theta_{j}\cap \theta_{k}|x_{i})].$$

We can use this function for the data compression of a sequence of labels or sentences.

Considering the semantic information of designs, decisions, and control plans, the G measure and the *R*(*G*) function are also useful. In these cases, the truth function *T*(*θ_j_*|*x*) becomes the DCF, *P*(*x*|*y*) becomes the realized distribution, information means control complexity or control amount, *G* means the effective control amount, and *R* means the control cost. To increase *G* = *I(X*; *Y_θ_*), we need to fix *T(θ_j_*|*x*) and optimize control to improve *P*(*x*|*y*). The *R*(*G*) function tells us that we can improve *P*(*y*|*x*) with the MMI iteration and increase *G* by amplifying *s*. In [32], the author improperly uses a different information formula (Equation (24) in [32]) for the above purpose. That formula seemingly only fits cases where the control results are continuous distributions. The author will further discuss this topic in another paper.

The G measure also has its limitations, which include:The G measure is only related to labels’ extensions, not to labels’ intensions. For example, “Old” means senility and closeness to death, whereas the G measure is only related to *T*(“Old”|*x*), which represents the extension of “Old”;The G measure is not enough for measuring the semantic information from fuzzy reasoning according to the context, although the author has discussed how to calculate compound propositions’ truth functions and fuzzy reasoning [32].

Therefore, we need more studies to measure more kinds of semantic information.

### 6.5. About the Importance of Semantic Information’s Studies

Shannon does not consider semantic information. He writes [34] (p. 3):
“These semantic aspects of communication are irrelevant to the engineering problem. The significance aspects is that the actual message is one selected from a set of possible messages. The system must be designed to operate for each possible selection, not just one which will actually be chosen since this is unknown at the time of design.”

However, the author of this paper does not think that Shannon opposed researching semantic information because, in the book co-written by Shannon and Weaver, Weaver [34] (pp. 95–97) initiated the study of semantic information. If we extend Shannon’s information theory for machine learning, which frequently deals with the selection of messages, we must consider semantic information.

Section 4.2 reveals that distortion is related to truth, truth is related to semantic meaning, and hence, the rate-distortion function is related to semantic information. Equation (31) indicates that the rate-distortion function *R*(*D*) can be expressed as semantic mutual information *I*(*Y*; *X_θ_*).

With machine learning’s developments, semantic information theory is becoming more important. From the perspective of the semantic information G theory, the EM algorithm for mixture models, where the E-step uses a formula like Equation (39), can be explained by the mutual matching of the semantic channel and the Shannon channel [40], so that two kinds of information are approximately equal. In the Restricted Boltzmann Machine (RBM), the Softmax function with the NEF and the partition function is used as the learning function [7]. Some researchers have discussed the similarity between the RBM and the EM algorithm [46]. It seems that we can also explain the RBM by the two channels’ mutual matching. We need more studies for the G theory’s applications to machine learning with neural networks.

## 7. Conclusions

In the introduction section, we raised three questions:Why does the Bayes-like formula (see Equation (2)) with NEFs and partition functions widely exist in the rate-distortion theory, statistical mechanics, the maximum entropy method, and machine learning?Can we combine machine learning (for the distortion function) and data compression (with the rate-distortion function)?Can we use the rate-distortion function or similar functions for semantic compression?

Using the semantic information G measure based on the P-T probability framework, we have explained that the NEFs are truth functions, the partition functions are logical probabilities, and the Bayes-like formula is a semantic Bayes’ formula. We have also explained that the semantic mutual information formula can express MMI *R* (see Equation (35)); maximum entropy (in statistical mechanics or the maximum entropy method) is equal or proportional to extremely maximum entropy minus semantic mutual information (see Equations (41) and (47)).

We have shown that we can obtain truth functions from sampling distributions by machine learning (see Equations (23) and (25)). Furthermore, after setting the relationship between the truth function and the distortion function (see Section 4.2), we can replace the distortion function with log(1/truth_function). Therefore, we can combine machine learning (for the distortion function) and data compression.

Since truth functions represent labels’ semantic meanings [15], we can extend the rate-distortion function *R*(*D*) to the rate-truth function *R*(*Θ*) by replacing the average distortion $\bar{d}$ with fuzzy entropy *H*(*Y_θ_*|*X*) (see Section 4.3). We can also extend *R*(*D*) to the rate-verisimilitude function *R*(*G*) by replacing the upper limit *D* of average distortion with the lower limit *G* of semantic mutual information (see Section 4.4). We have used two examples to show how the iteration algorithm let the Shannon channel *P*(*y*|*x*) match the semantic channel *T*(*y*|*x*) under the semantic constraints to achieve MMI *R*. Example 1 reveals that the iteration algorithm can use the logical implication between labels to reduce mutual information. Example 2 indicates that it is easy to control the Shannon channel *P*(*y*|*x*) with the semantic channel *T*(*y*|*x*) for data reduction.

However, we also need to compress and recover text data (according to the context), image data, and speech data (according to the results of pattern recognition or species recognition). The above extension is useful, but it is far from sufficient, and so further studies are necessary.

## Figures and Tables

**Figure 1 entropy-23-01050-f001:**
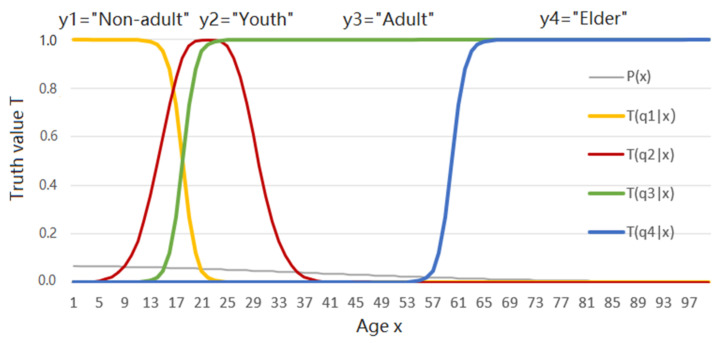
Four labels’ (fuzzy) truth functions about people’s ages. *T*(*y_j_*|*x*) denotes the truth function of proposition function *y_j_*(*x*) and *j* = 1, 2, 3, 4, as the constraint condition.

**Figure 2 entropy-23-01050-f002:**
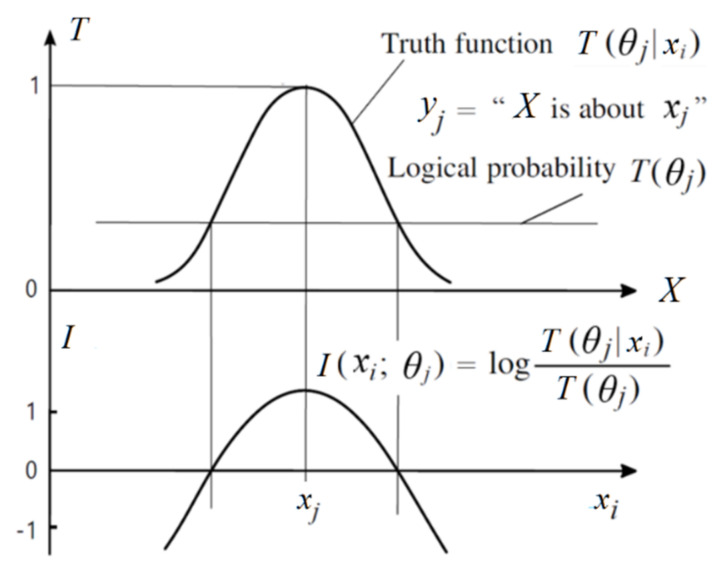
The semantic information conveyed by *y_j_* about *x_i_*.

**Figure 3 entropy-23-01050-f003:**
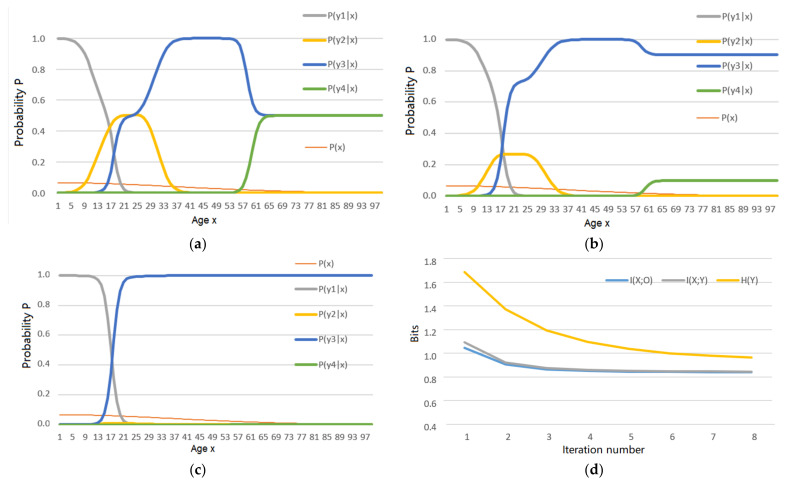
*P*(*y*|*x*) and *I*(*X*; *Y*) change in the iterative process for *R*(*Θ*). (**a**) *P*(*y*|*x*) after the first iteration (this *P*(*y*|*x*) also results in maximum entropies *H*(*Y*|*X*) and *H*(*X*, *Y*)); (**b**) *P*(*y*|*x*) after the second iteration; (**c**) *P*(*y*|*x*) after eight iterations; (**d**) *I*(*X*; *Y*), *I*(*Y*; *X_θ_*), and *H*(*Y*) change in the iterative process.

**Figure 4 entropy-23-01050-f004:**
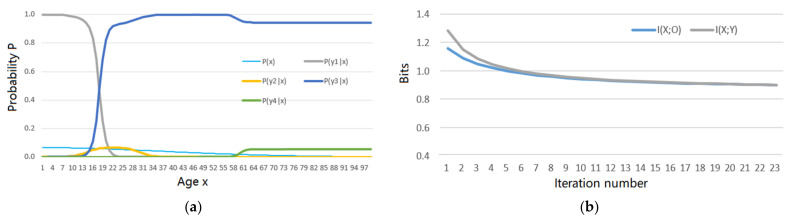
The iterative results for the *R*(*G*) function. (**a**) The convergent Shannon channel *P*(*y*|*x*); (**b**) *I*(*X*; *Y_θ_*) and *I*(*X*; *Y*) = *I*(*Y*; *X_θ_*) change in the iterative process.

**Figure 5 entropy-23-01050-f005:**
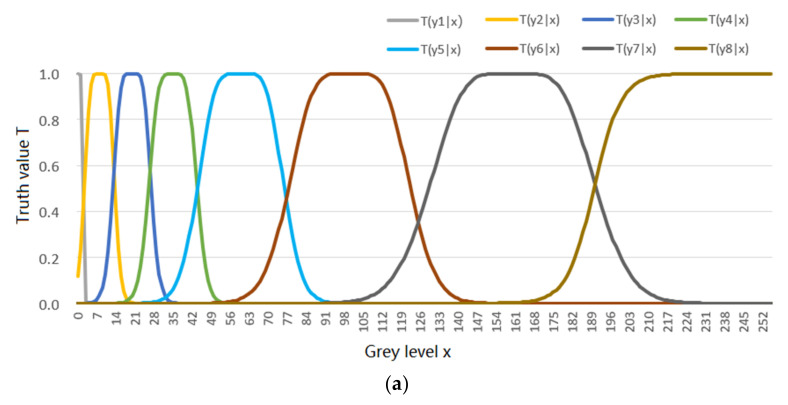
The results of Example 2. (**a**) Eight truth functions or the semantic channel *T*(*y*|*x*); (**b**) The convergent Shannon channel *P*(*y*|*x*); (**c**) *I*(*X*; *Y_θ_*), *I*(*X*; *Y*) and *H*(*Y*) change in the iterative process.

## Data Availability

Not applicable.

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
