# Peer review of "Using the Semantic Information G Measure to Explain and Extend Rate-Distortion Functions and Maximum Entropy Distributions"

_entropy, 2021, doi:10.3390/e23081050_

Round 1

Author Response

Please see the pdf file.

Reviewer 2 Report

This manuscript explores semantic compression by building on the author’s previous work on semantic information G theory, and associating truth functions and distortion functions. The author discusses the limitations of the rate distortion function when semantic compression is involved. Examples are presented which show how truth functions can be used as constraints in MMI iterations to obtain the convergent Shannon channel.

The manuscript is well-written and argued. I have a few comments with respect to the presentation.

  1. A Conclusions section where the principal results of the manuscript are detailed, would be helpful for the reader.
  2. In Figure 1, it is difficult to discern between the blue colors used for the second (y_2) and fourth (y_4) truth functions. I would recommend replacing one with a different color. This same comment for y_3 and y_5 in Figure 6 (a) and y_5, y_7 in Figure 6(b).

In Figure 6 (b), the label on the y-axis is ‘conditional probability’, whereas in Figure 5 (a), it is ‘probability’. Figures 4 (a), (b), (c ) have no labels on the y-axis.

Figure 6 (a) has a y-axis with label ‘truth value’ while the y-axis of Figure 1 has no label.

In the plots of the iterations in Figures 4 (d) and 6 (c ), the y-axis is labelled with ‘bits’. There is no label in Figure 5 (b).

The x-axes in Figures 6 (c ) and 4 (d) are labelled with ‘iterations’ while in Figure 5 (b) ‘iteration number’ is used.

In Figure 1 and in Figures 4 (a)-(c ), the x-axis is labelled ‘Age x’ while in Figure 5 (c ) it is ‘x age’.

Would the plots of the iterations in Figures 4 (d) and 5 (b) be better presented if the values used on the y-axis were between 0.8-1.8 and 0.8-1.4 respectively? This would be then consistent in presentation with that of Figure 6 (c ). The differences and behavior of the curves would be better appreciated. 

3. I could not understand the discussion of Figure 2 on lines 273-274, ‘Figure 2 indicates that the less the logical probability is, the more information there is;’. Isn’t the logical probability a constant value as indicated by the horizontal grey line in Figure 2?

4. The definitions of the truth functions presented in Appendix B (the second Appendix B should be Appendix C?). Are these functions that are generally accepted for use? Have the iterative methods been tested with other functions (and parameters) with similar results? The author could comment on this for the generalist reader. 

Author Response

Please see the pdf file.

Reviewer 3 Report

First of all, the manuscript lacks a readable introduction and a
statement of the results which would be accessible enough. It seems to
touch many different topics like the maximum entropy principle, the
rate-distortion function, and some sort of semantics but it does not
give in-depth motivations and overviews for its developments in plain
language. The algebraic operations look quite trivial. For this
reason, I am not able to claim that the actual contribution of the
author is significant enough. I am concerned particularly about the
formal distinction between statistical and logical probability and the
concept of semantic information, which hardly have been explained and
rooted in some common-sense intuitions. I estimate that the necessary
work to make this paper understandable (which does not guarantee its
publication) would be adding about ten pages of comments in plain
language. For this reason I recommend rejecting this paper in its
present form.

Author Response

Please see the pdf file.

Round 2

Reviewer 2 Report

The author has revised the manuscript in accord with the previous comments. I recommend publication.

Author Response

Dear Reviewer 2:

Thank you for understanding and support!  My paper is challenging. I am lucky to get your support.

Best wishes!

Chenguang Lu

Reviewer 3 Report

Seeing other reviews, I have decided to be more lenient in my second
round review. Given the short deadline, I am not able read the
manuscript thoroughly enough and fully extract and appreciate its
original thoughts. As I grasp upon superficial browsing, a certain
general idea of the author is to introduce semantics into information
theory. The author should be made aware that a few other researchers
tried to bring together semantics and information theory, such as:

Bar-Hillel, Y.; Carnap, R. An Outline of a Theory of Semantic
Information. In Language and Information: Selected Essays on Their
Theory and Application; Reading: Addison-Wesley, 1964; pp. 221–274.

Debowski, L. Information Theory Meets Power Laws: Stochastic Processes
and Language Models; New York: Wiley & Sons, 2021.

The general problem of combining semantics and information theory is
how to conceptualize meanings in information theory. There is no
essential problem when we treat meanings as discrete symbols from a
finite alphabet. Essentially, finite alphabets are at the core of
information-theoretic formalism, so adding a semantic interpretation
to individual discrete symbols does not introduce a substantial
intellectual novelty. What may be more interesting, is to see how
combinations of symbols may acquire a compositional meaning, whether
the number of compositional meanings can be unbounded, etc. The
meaningfulness of a text can be also ambiguously understood as one of
the following properties: (1) a consistent description of some
extratextual reality, (2) a coherent narration, full of internal
connections, (3) an achievement of some predefined goal, such as a
particular mental state of the reader. I suppose that to properly
understand such phenomena we need to inspect also the concepts of
algorithmic randomness and Kolmogorov complexity.

My first negative review came as a result of seeing that the author
had not even thought about such problems and set his goals much less
ambitiously. Now I think that the present manuscript can be accepted
for publication after some minor amendments but the author should be
strongly encouraged to think more seriously about semantics and
information theory in the future.

Author Response

Please see the letter in the pdf file.

This manuscript is a resubmission of an earlier submission. The following is a list of the peer review reports and author responses from that submission.

Round 1

Reviewer 2 Report

In the manuscript "Using Semantic Information G Measure to Explain and Extend Rate-distortion Functions and Maximum Entropy Distributions", the author proposes to use semantic information G theory to reinterpret negative exponential functions as truth functions and partition functions as logical probabilities and draws a couple of conclusions from this reinterpretation. They present examples to support the utility of this reinterpretation.

While the idea of reinterpreting well known terms of information theory in a new perspective, I fail to see sufficient novelty in the manuscript. The semantic information G theory was proposed by the author in a series of publications in the 1990s. This framework is presented in section 3. Section 4 then starts with an interpretation of the MMI and rate-distortion function and distortion function in terms of semantic information G theory. The rate-of-limiting error function (line 357 ff.) was already defined in [28]. Here, the author just clarifies the relation between the distortion function and the truth function. Similarly, the rate-verisimilitude function R(G) of Section 4.4 was explored in [28] and [29] as well.

Regarding the experimental results, Section 5.1 describes a simple constructed toy example with fuzzy age classes. Due to its simplicity, this example fails to demonstrate practical utility.

Section 5.2 describes a comparison of the E3M algorithm to the classic expectation maximization algorithm. First, the E3M algorithm is a particular instance of the CM-EM algorithm, which was already discussed in [29] and [37]. Second, regarding the comparison of the E3M algorithm vs the EM: E3M needs 240 iterations while EM needs 340. However, in the E3M algorithm the E-step and the M1-step are applied multiple times per iteration. A comparison of the number of required iterations is therefore meaningless. The running time improvement mentioned in line 585 is a better comparison but might suffer from implementation details. No details are provided. The author merely points to reference [37] for details.

Overall, the manuscript proposes minor reinterpretations based on a decades old framework and presents unconvincing experimental results.

Minor comments:

Eq. 2: The distortion function d is not introduced.

Line 63: An explanation for why to minimize would make the manuscript for accessible to non-experts.

Line 64: The statement that the "rate-distortion cannot work well" is not clear and should be made more precise.

Eq. 9: The use of average vs expectation (which I suppose 'E' refers to) is not rigorous. E is not explained.

Line 338: j should be subscript.

Figures 4d, 5b: There are many useless decimal places on the y axis ticks.